# Phenotypic plasticity as an adaptation to a functional trade-off

Xiao Yi[1][*][†], Antony M Dean[1,2][*]

[1]Department of Ecology, Evolution and Behavior, University of Minnesota, St. Paul, United States; [2]Laboratory of Microbial Evolution, College of Ecology and Evolution, Guangzhou, Peoples Republic of China

**Abstract** We report the evolution of a phenotypically plastic behavior that circumvents the hardwired trade-off that exists when resources are partitioned between growth and motility in *Escherichia coli*. We propagated cultures in a cyclical environment, alternating between growth up to carrying capacity and selection for chemotaxis. Initial adaptations boosted overall swimming speed at the expense of growth. The effect of the trade-off was subsequently eased through a change in behavior; while individual cells reduced motility during exponential growth, the faction of the population that was motile increased as the carrying capacity was approached. This plastic behavior was produced by a single amino acid replacement in FliA, a regulatory protein central to the chemotaxis network. Our results illustrate how phenotypic plasticity potentiates evolvability by opening up new regions of the adaptive landscape.

**\*For correspondence:**
xiaoyi8607@gmail.com (XY);
deanx024@umn.edu (AMD)

**Present address:**
[†]BioTechnology Institute, University of Minnesota, Saint Paul, United States

**Competing interests:** The authors declare that no competing interests exist.

## Introduction

Trade-offs arise when finite resources are allocated to different traits such that an increase in one causes a decrease in another. In variable environments trade-offs can be imposed consecutively, with different traits favored at different times. There are now two possible adaptive responses: (i) simultaneously optimize the traits or (ii) change the traits dynamically to match the prevailing conditions, a phenomenon called phenotypic plasticity. Theoretical models predict that, provided the genetic variation exists to generate a suitably plastic response, the second should be favored because it reduces the costs imposed by the trade-offs (*De Jong, 1993*; King and Roff, 2010; *Malausa et al., 2005*; *Murren et al., 2015*; *Roff, 2001*; *Scheiner, 1993*; *van Noordwijk and de Jong, 1986*; *Worley et al., 2003*). For example, changes in insect wing morphology with varying environmental conditions, such as temperature and food quality, have been attributed to the trade-off between flight capability and reproduction (*Harrison, 1980*; *Zera et al., 1997*). Wings develop fully only in stressful environments, allowing insects to escape adversity while minimizing costs to reproduction in benign environments (*Harrison, 1980*).

Empirical examples of both phenotypic plasticity and trade-offs abound, but rarely has phenotypic plasticity been proved to be an adaptive response to trade-offs (King and Roff, 2010). Nor have the genetic architectures underlying phenotypic plasticity been identified (*Murren et al., 2015*). These difficulties are due to the lack of a simple tractable experimental system.

Here, we use *Escherichia coli* to explore a trade-off between two traits, growth rate and chemotaxis – the ability to move up a concentration gradient of nutrients. By monitoring phenotypic evolution in a well-defined selective environment, we can show how these traits are first simultaneously optimized before an adaptively plastic response evolves that gains access to a new region of the adaptive landscape.

Extensive studies from molecular biology and physiology have established that both growth and chemotaxis are energetically expensive and critically important to fitness (*Freter and O'Brien, 1981*;

*Macnab, 1996*). Given bacterial metabolic rates are limited (*Makarieva et al., 2008*), there must be a trade-off between growth and chemotaxis. To test this, five populations of wildtype *E. coli* were propagated in a serial transfer regime that alternated between competitive growth in batch culture and capillary selection for chemotaxis (*Figure 1a*).

Our experimental system defines an adaptive landscape in which selection for limited resources during growth is modeled using Lotka-Volterra equations (*Gotelli, 1998*),

$$\frac{dm}{dt} = r_m m (1 - m - w)$$
$$\frac{dw}{dt} = r_w w (1 - m - w)$$

where $m$ and $w$ are the population sizes for the mutant and wildtype normalized to their common carrying capacity, and $r_m$ and $r_w$ are their respective intrinsic rates of growth. The selection after 11.5 hr of growth, defined as $s_g = Log_e(m_{11.5}/w_{11.5}) - Log_e(m_0/w_0)$, is determined by numerical integration of the Lotka-Volterra equations. The selection after chemotaxis is $s_c = Log_e(m_{12}/w_{12}) - Log_e(m_{11.5}/w_{11.5}) = 0.5\,(c_m - c_w)$ in which $c_m - c_w$ is the difference in chemotactic ability (this difference is equivalent to the difference in growth rates $(r_m - r_w)$ as populations increase exponentially in both situations [*Adler, 1973*]). There are approximately 11 population doublings (generations) per growth cycle (effectively a 2048-fold dilution into fresh growth medium at the beginning of each growth cycle). Relative fitness of the mutant to wildtype is

$$w = (s_g + s_c)/12$$

Plotting $w$ as a function of the differences in the rates of growth $(r_m - r_w)$ and chemotaxis $(c_m - c_w)$ produces an adaptive surface (*Figure 1b*) whose curvature is attributable to the deceleration in

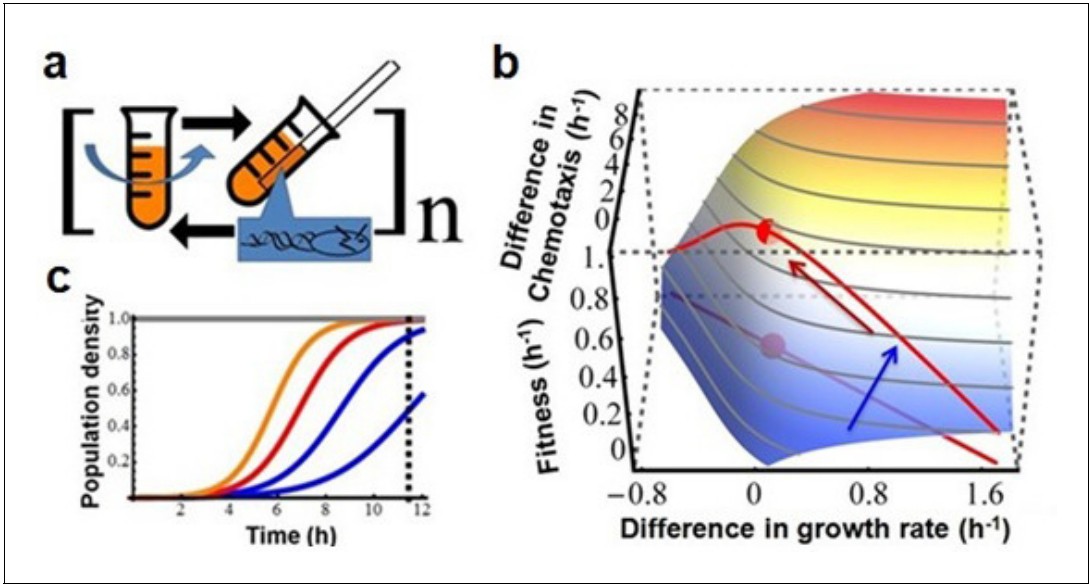

**Figure 1.** Testing the existence of a trade-off between growth rate and chemotaxis using experimental evolution. (a) Experimental design. Bacteria are grown in a batch culture of rich medium for 11.5 hr to late exponential phase (*Figure 2b*, dashed curve). A capillary carrying fresh medium is lowered into the culture (after wash and dilution, see Materials and methods) for 30 min to attract chemotactically active cells. Cells thus collected are used to inoculate another batch culture. The cycle is repeated 150 times (approximately1650 generations). (b) Predicted adaptive landscape. The experimental system defines an adaptive landscape that can be described mathematically from first principles. Relative fitness ($w$) plotted as a function of the differences in growth rates $(r_m - r_w)$ and chemotactic abilities $(c_m - c_w)$, is determined a priori by the experimental conditions and is robust to violations of assumptions (Appendix).The trade-off, who's position must be determined empirically, is absent when a population evolves along a positive diagonal (blue arrow) and present when a population evolves along a negative diagonal (red arrow) (*Agrawal et al., 2010*). The red dot marks maximum fitness on the Pareto front of the trade-off. (c) Curvature in the landscape is produced by the population density at the point of capillary selection (dashed line) becoming less dependent on growth rate as the latter is increased from low (blue) to high (red and orange). In other words, as growth rate increases, its contribution to fitness diminishes due to the sigmoid nature of logistic growth.

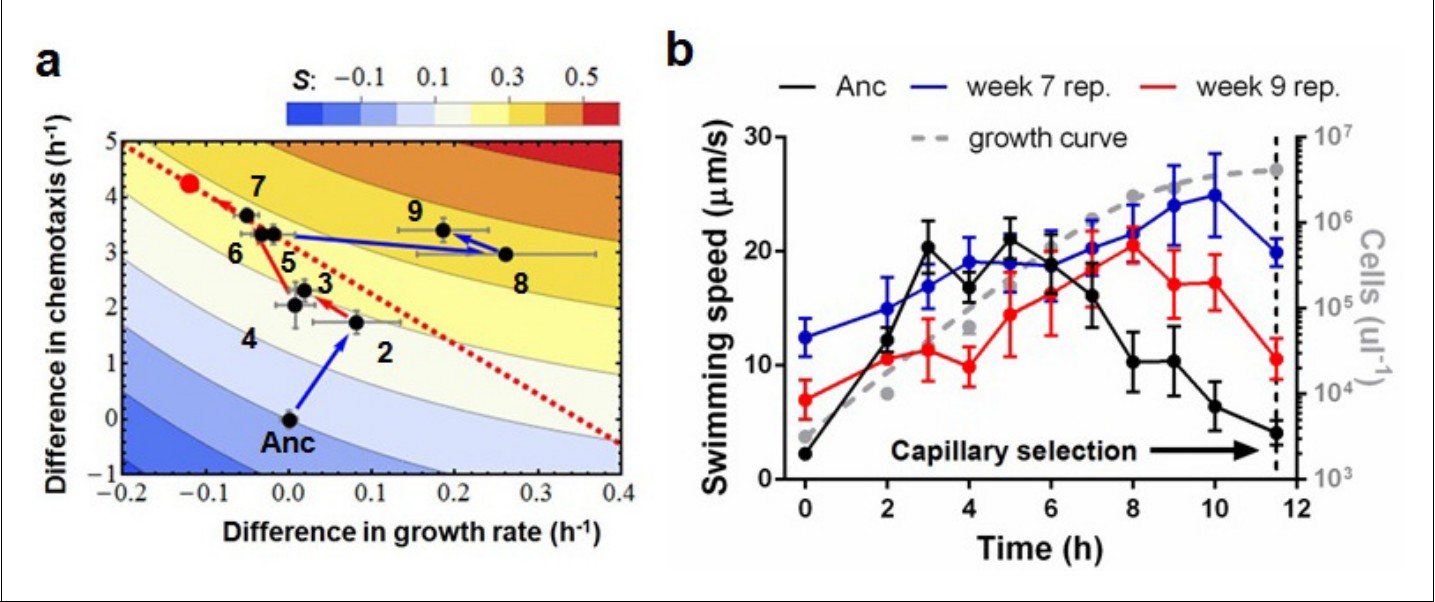

**Figure 2.** Phenotypic plasticity in the face of a trade-off between growth rate and chemotaxis. (a) An evolutionary trajectory through the known adaptive landscape. Numbers indicate the stage of evolution in weeks. The dashed line denotes the empirically determined Pareto front and the red dot denotes its fitness maximum. Contours mark fitness isoclines. Each point represents the average of six clones randomly picked from the evolving population. The large standard errors noticeable at week 8 are a consequence of transient polymorphisms as new fitter mutants sweep into the population. (b) Swimming speeds of representative clones from different stages. Each point represents the mean of five replicate populations, each of which is the mean of ~400 cells. Error bars represent standard deviations. The dashed curve indicates a typical growth curve with its vertical axis on the right. After six hours, growth rate diminish and the populations start transitioning from exponential to stationary phase. Each point is the average taken from three replicate populations. Error bar are standard deviations.

The following figure supplements are available for figure 2:

**Figure supplement 1.** The overall evolutionary trajectory of all five populations.

**Figure supplement 2.** Growth rate evolves 11.6% faster within 3 weeks (c.a. 460 generations) in this control experiment without capillary selection.

**Figure supplement 3.** Empirical Pareto front and theoretical fitness optimum.

**Figure supplement 4.** Correlation between swimming speed and chemotactic ability.

**Figure supplement 5.** Test of *E. coli* chemotaxis at the late stage.

**Figure supplement 6.** Logistic growth curves fitted to data from isolated clones.

**Figure supplement 7.** The impact of *r* and *k* on competition at the end of batch growth, *f*, **a**, and its derivatives, **b**.

growth rates as the cultures approach the carrying capacity (*Figure 1c*). Projecting a hypothetical Pareto front (representing the limit where trade-offs force phenotypes to become tightly negatively correlated) onto the surface produces a maximum. The existence of the trade-off and the location of the Pareto front must be determined empirically. Cultures evolving towards the adaptive peak (red dot, *Figure 1b*) show that natural selection maximizes fitness in the face of a trade-off.

Evolutionary trajectories across this landscape are characterized by three phases. *Figure 2a* depicts a typical trajectory (see *Figure 2—figure supplement 1* for additional data). During the first phase (weeks 1 and 2) substantial increases in chemotactic ability and fitness occur (ANOVA, p<0.0001; p<0.0001); few isolates have changed growth rates. Therefore no trade-off is present. Control experiments without chemotaxis select for improved growth rates (p=0.0022, *Figure 2—figure supplement 2*). The selective gradient (*Lynch, 1998*) (a measure of selective strength)

along the growth axis is much steeper than along the chemotaxis axis (0.603 vs. 0.087, Materials and methods). Therefore, adaptation in the early phase is dominated by chemotaxis mutations of large phenotypic effect.

The second phase (weeks 2 to 7) is characterized by adaptation to a trade-off. Growth rates decline(p=0.029) as chemotaxis continues to improve (p=0.0018). Regressing growth rate against chemotaxis (data from weeks 5–7, *Figure 2—figure supplement 3*) reveals a Pareto front (dashed line, *Figure 2a*) that almost parallels the fitness isoclines. This explains why fitness improves at a slower rate (p=0.0009). By week 6 this evolving population (like the other four) had approached the predicted fitness optimum (red dot, *Figure 2a*).

Most previous studies of trade-offs have faced difficulties in relating phenotypes to fitness (*Agrawal et al., 2010*; *Allouche et al., 2012*; *Fonseca-Azevedo and Herculano-Houzel, 2012*; *Keen, 2014*; *Keller et al., 2014*; *King et al., 2004*; *Mole and Zera, 1993*; *Phan and Ferenci, 2013*; *Shoval et al., 2012*). In our adaptive landscape the relationships between phenotypes and fitness were defined a priori. Independent regressions from different experimental populations produce similar Pareto fronts (*Figure 2—figure supplement 1*), with all populations evolving close to the predicted optimum (*Figure 2—figure supplement 1*). This establishes a hard-wired trade-off between chemotaxis and growth rate attributable to resource/energy limitation (*DeLong et al., 2010*). The match between the prediction and experimental realizations shows that the interplay of the selective forces is understood and that adaptation is predictable at the phenotypic level.

Week 8 in the third phase saw populations break through the Pareto front with large increases in growth rate (p=0.0068, comparing weeks 7 and 9) while maintaining strong chemotaxis (p=0.77). To understand how the hard trade-off forced by partitioning energy between chemotaxis and growth was overcome we isolated individual clones from the three phases and determined their swimming speed over the course of growth in batch culture. Three patterns were evident (*Figure 2b*): (1) the ancestor's swimming speed increases during rapid growth and then declines as the carrying capacity is approached, (2) clones from the early and the middle periods swim faster than the ancestor nearly the whole time, the difference becoming more marked during late growth, and (3) clones that broke through the Pareto front reduce swimming speed during rapid growth and increase swimming speed as the carrying capacity is approached.

Chemotactic ability is strongly correlated with swimming speed (*Figure 2—figure supplement 4*). One adaptation evident in isolates from the first and second phases simply increases swimming speed throughout the growth cycle. This requires diverting resources away from growth, which slows. Genomic sequencing of a phase 2 clone identified two relevant mutations (*Table 1*):a non-synonymous substitution in *yahA* and another substitution in the promoter region of *yegE*. Both genes encode enzymes that metabolize(3′-5′)-cyclic dimeric guanosine monophosphate (c-di-GMP) (*Claret et al., 2007*; *Pesavento et al., 2008*), a second messenger that negatively modulates flagellar activity in *E. coli* (*Boehm et al., 2010*; *Pesavento et al., 2008*). We replaced the *E. coli lac* operon with a synthetic construct in which expression of GFP is post-transcriptionally modulated by the riboswitch, Vc2, which senses c-di-GMP (*Sudarsan et al., 2008*). As expected, GFP expression shows that the phase 2 isolate has lower intracellular c-di-GMP concentrations at the beginning and end of the growth cycle when its swimming speed is higher than the ancestor (*Figure 3a*).

Isolates from the third phase reduced motility during growth only to boost it when approaching carrying capacity in preparation for chemotaxis selection. Genomic sequencing of a phase 3 clone identified a non-synonymous substitution that replaced a conserved arginine with a tryptophan at site 220 in the DNA binding domain of FliA, a transcription factor essential for chemotaxis and motility (*Claret et al., 2007*). Structural modeling (*Figure 3—figure supplement 1*) predicts that R220W reduces the electrostatic interaction with, and eliminates a hydrogen bond to, the backbone phosphate of DNA.

To determine if the R220W mutation reduces flagellar operon transcriptional activity we replaced the *E. coli lac* operon with a synthetic construct in which GFP expression is driven from the *fliA*-dependent *fliC* promoter. We constructed four strains: two genetic backgrounds (ancestor and phase 3) each with either of two *fliA* alleles (wildtype and mutant). As expected, FliA[R220W] reduced overall GFP expression in both backgrounds, with the loss of peak expression causing a reduction in the swimming speed of *individual* cells early in the cycle (*Figure 3b*).

Despite the overall reduction in transcription from *fliC*, FliA[R220W] increases average swimming speed late in the growth cycle (*Figure 3c,d*). This is achieved by boosting the fraction of cells in the

**Table 1.** Complete mutations of representative isolates from week 7 and week 9. Note that there is no overlap in mutations between the two isolates.

| | Nucleotide change | Animo acid substitutionor genomic context | Genes | Phenotypes |
|---|---|---|---|---|
| Week 7 isolate | G→T | G167C (GGT→TGT) | yahA → | c-di-GMP-specific phosphodiesterase |
| | G→T | L356L (CTC→CTA) | dcp ← | dipeptidylcarboxypeptidase II |
| | repeat_region (−) +4 bp | intergenic (−209/−52) | udk ← / → yegE | uridine/cytidine kinase/diguanylatecyclase |
| | repeat_region (+) +4 bp | intergenic (+113/−115) | yqaD → / → CsiD | orf, hypothetical protein/orf, hypothetical protein |
| | repeat_region (−) +9 bp | coding (1504– 1512/2247 nt) | ptsP ← | PTS system, enzyme I, transcriptional regulator (with NPR and NTR proteins) |
| Week 9 isolate | Δ5579 bp | | [gapC]–[ydcJ] | [gapC], cybB, ydcA, hokB, mokB, trg, ydcI, [ydcJ] |
| | G→A | R220W (CGG→TGG) | fliA ← | flagellar biosynthesis; alternative sigma factor 28; regulation of flagellar operons |
| | C→T | E82K (GAA→AAA) | rpsC ← | 30S ribosomal subunit protein S3 |
| | repeat_region (+) +5 bp | intergenic (+86/−36) | rbsB → / → rbsK | D-ribose periplasmic binding protein/ribokinase |

*population* that are motile (*Figure 3e*). Note that the boost is bigger in the ancestral than the phase 3 background where three additional mutations (*Table 1*) also raise the fraction of motile cells. Hence, reduced motility early in the growth cycle and increased motility late in the growth cycle are caused by a single pleiotropic mutation, R220W. This mutation is expected to increase fitness in the face of the trade-off.

Directly measuring the two fitness components of strains carrying wildtype or mutant versions of FliA (*Figure 3f*) shows thatFliA$^{R220W}$ elevated fitness in two ways: (1) increasing growth rates by reducing flagellar transcription and the associated energetic costs to individual cells, and (2) increasing the fraction of the population that is motile during chemotaxis selection. Hence, a single mutation, with pleiotropic effects at the levels of individuals and of populations, produces the plastic behavior needed to adapt to a dynamic environment.

The first gains in fitness were achieved by a significant boost in chemotactic ability, with motility becoming less plastic (compared to the ancestor) across the growth cycle. Further improvements in fitness were limited by the growth-chemotaxis trade-off. Eventually, however, a new pattern of motility evolved across the growth cycle that mitigated the trade-off. These observations illustrate the adaptive value of phenotypic plasticity in variable environments whenever fitness components are bounded by trade-offs. FliA regulates the concerted expression of more than 40 chemotaxis genes (*Claret et al., 2007*). Characterizing the adaptive *fliA* mutant has revealed how a novel adaptive plastic behavior readily evolves by tinkering with a node at the center of an existing gene network. Delineating the mechanistic basis of a behavioral adaptation, our approach serves to complement those many studies of phenotypic plasticity that have focused on genome wide associations (*Ghalambor et al., 2015*; *Gompert et al., 2014*).

There are three different ways adaptation can resolve constraints caused by functional trade-offs. First, gain-of-function mutations can change the constraint itself. For example, *E. coli* can evolve the ability to use citrate as a carbon source under aerobic conditions (*Hall, 1982*), thereby eliminating a metabolic constraint. Second, adaptation can optimize traits to bring populations to an adaptive peak. Echoing previous work on adaptive landscapes where trade-offs serve as boundary conditions to provide strong predictive power (*Dekel and Alon, 2005*; *Ibarra et al., 2002*; *Poelwijk et al., 2011*), we observed that growth rate and chemotaxis were initially optimized to bring populations close to the optimum on the Pareto front. Third, in a variable environment the costs associated with a trade-off can be mitigated by a phenotypically plastic response. The unanticipated escape from the optimum illustrates how the emergence of a new behavior can reduce the costs of a trade-off to gain access to virgin regions of an adaptive landscape. Recently, another group found that a trade-off between different environments facilitated adaptation (*De Vos et al., 2015*).This adds to our

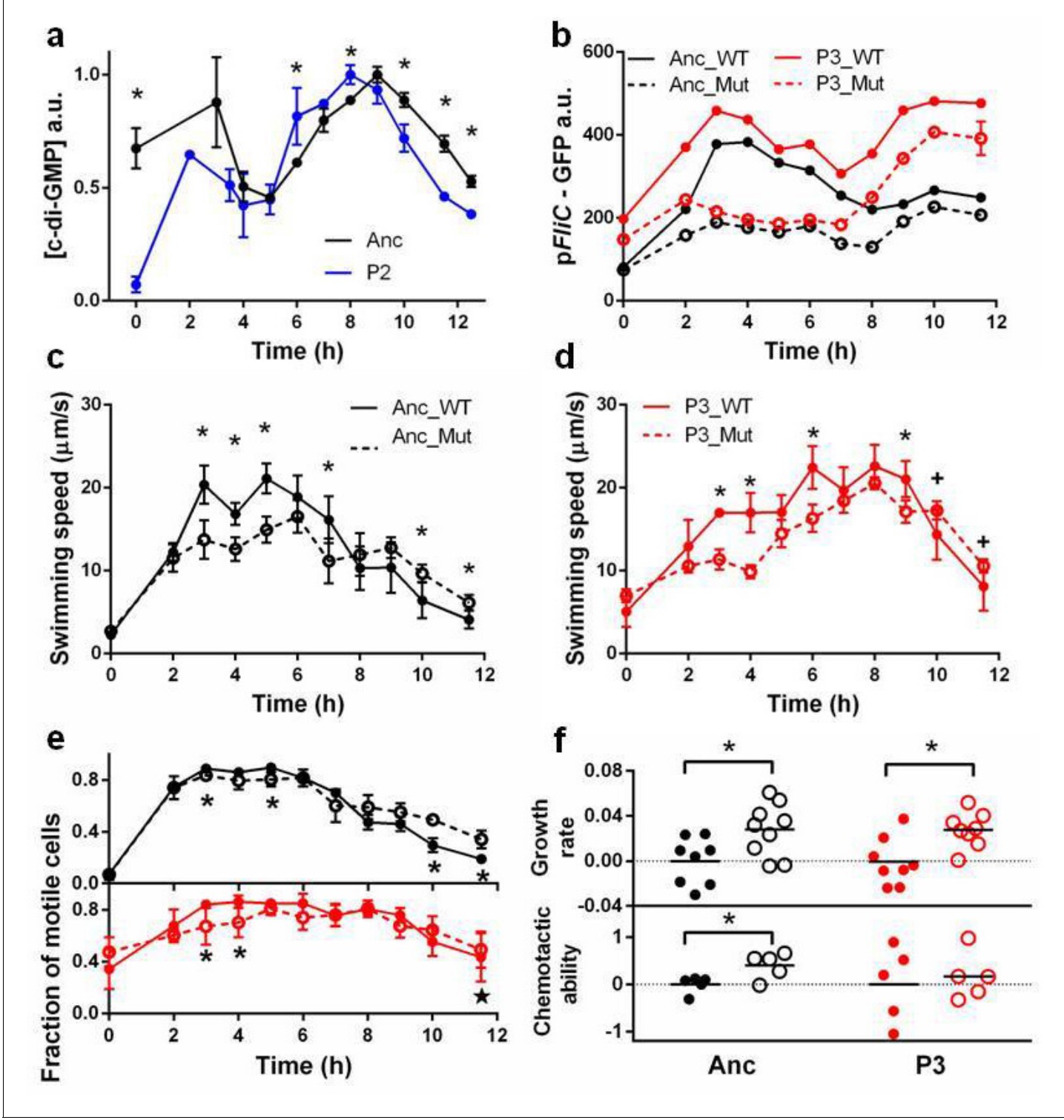

**Figure 3.** Molecular, physiological and fitness impacts of the adaptive mutations. (**a**) Intracellular concentrations of c-di-GMP in the ancestor (Anc) and phase 2 isolate (P2) as measured by Vc2 riboswitch (*Sudarsan et al., 2008*) expressed GFP. (**b**) GFP expression driven from an ectopic *fliC* promoter in the ancestor and in the phase 3 isolate (P3). Each point represents the mean of three replicates. (**c**) and (**d**). Swimming speed of wildtype *fliA* (solid curve) and mutant *fliA* (dashed curve) in the ancestral (black) or phase 3 (red) genetic backgrounds. Each point represents the mean of five replicate populations, each of which is the mean of ~400 cells. Error bars represent standard deviations. (**e**) Fraction of motile cells within isogenic populations. Legend and data symbols follow that of **c** and **d**. (**f**) Effect of the *fliA* mutation on growth rate and chemotaxis. Ancestral background (black), phase 3 background (red), wildtype *fliA* (filled circles) and mutant *fliA* (open circles). Values are normalized to mean values of wildtype *fliA*. Bar indicates mean.
*Figure 3 continued on next page*

*Figure 3 continued*

For all panels, * indicates statistical significance at least of 0.05 with three to five replicate populations; ★, with nine replicate populations; + , with 200 cells randomly pooled from the replicate populations for comparison.
The following figure supplements are available for figure 3:

**Figure supplement 1.** Comparison between wildtype and mutant sigma factor F at the DNA binding domain.
**Figure supplement 2.** Statistical structure of swimming speed within isogenic populations and its impact on chemotaxis.
**Figure supplement 3.** Comparison of measurements taken in spent medium and in chemotactic buffer.

speculation that, besides phenotypic plasticity, many other adaptive novel traits have their origins in tradeoffs.

## Materials and methods

### Strains and media

*E. coli* strain MG1655 (Coli Genetic Stock Center, Yale) was propagated in tryptone broth (TB; 10 g $NaCl_2$, 5 g tryptone per liter; Fisher BioReagents). Chemotactic competition and swimming speed were determined in washing buffer (7 g $K_2HPO_4$, 2 g $KH_2PO_4$ per liter, 0.1 mM EDTA, Sigma). Evolved clones were isolated by streaking on LB plates (10 g NaCl2, 5 g yeast extract 5 g, 10 g tryptone, 15 g agar per liter; Fisher BioReagents). Genetic engineering followed standard protocols as detailed in supplemental file 1.

### Experimental evolution

*E. coli* cells were selected for growth rate and chemotaxis in a fast-paced cyclical environment. From a single colony, the ancestral strain was grown overnight at 30°C to full density and diluted 1000 times into each of five 18 mm sterile glass tubes (Pyrex) containing one ml TB. The five cultures were treated the same way. They were incubated at 30°C and shaken at 250 rpm. 11.5 hr after inoculation, 50 µl of each culture was diluted into 250 µl of washing buffer and centrifuged at 3000 g for 3 min. The supernatant was discarded and the pellet gently re-suspended in 300 µl of washing buffer to minimize flagellar damage. 180 µl of the suspension was transferred to a well in a sterile 96-well microplate (BD Falcon). Having loaded all five samples, the plate was raised on one side so that it became perpendicular to the bench surface. The liquid remained inside the wells due to surface tension. A glass micro-capillary (0.8 mm inner diameter × 75 mm, Drummond) was heat-melted at one end to seal the opening. After cooling to room temperature, the capillary was flamed over Bunsen burner briefly and the open end immediately submerged into TB. On cooling to room temperature the capillary absorbed ~5 µl of TB. Washing buffer was pipetted to rinse off carryover medium on the outer surface of the capillary. The open end was submerged into the liquid in the well and the whole system was incubated at 30°C for chemotactic competition to occur. After 30 min, the capillary was removed and rinsed with washing buffer to remove cells on the outer wall. To initiate the next cycle of growth and chemotaxis, the liquid and cells in the capillary were pushed into a glass tube of 1 ml of fresh TB medium by flaming the capillary briefly. This cycle of growth and chemotaxis was repeated 150 times (11 weeks). Each week 1 ml of culture was taken from each tube and diluted with 80% sterile glycerol to the final glycerol concentration of 16%. The diluted cultures were stored at −80°C as archives. For each archived culture, 6 single strains were randomly picked from single colonies streaked on agar plate. Each strain was characterized for growth rate and chemotactic ability.

### Measuring growth rates and chemotactic abilities

Overnight cultures were diluted 1000-fold into 10 mls of fresh tryptone broth and incubated at 30°C with vigorous shaking. Cell counts were determined by flow cytometry at 3, 4 and 5 hr after inoculation. Growth rates were determined as the slope of linear regression line of logarithmic transformed

counts against time. At 11.5 hr, samples from both cultures were centrifuged at low speed and re-suspended in six times the volume of washing buffer. The mutant and the ancestor (carrying a selectively neutral *fhuA* marker that confers resistance to the bacteriophage T5) were mixed in a 1:1 ratio in 180 μl. A micro-capillary carrying ~5 μl tryptone broth was lowered to the mixture, and the system incubated at 30°C for 30 min. Cell counts from pre- and post-incubation (inside the capillary) were determined by flow cytometry (details in supplemental file 1). The difference between the logarithm-transformed ratios, before and after incubation, is a measure of chemotaxis of a mutant relative to the ancestor.

## Measuring swimming speed

Culture samples, diluted in washing buffer or concentrated by low-speed centrifugation according to cell density, were placed on microscope slides. Five second video clips were taken from different fields using a microscope (Olympus IX70 Inverted Fluorescence Microscope, SPOT Flex camera). The swimming speeds of individual cells were determined for the trajectories extracted using Image-Pro Plus 6.0 software.

## Calculation of selective gradients

We studied the selective gradients along the two functional dimensions. A grid of values for c and r was generated that covered the parameter space traveled by the evolving populations: for c from −1 to 8 with increment size 0.2 and for r from −0.2 to 0.3 with increment size 0.025. The fitness, $w_{ij}$, at each point, $(c_{ij}, r_{ij})$, was calculated, where i and j were the indexes for the simulated values of c and r respectively. The surface made of these points is smooth and approximates a plane. Therefore, the overall selective gradients can be approximated by averaging the gradients as follows:

$$\frac{\partial w}{\partial c} = \frac{1}{0.2} \sum_j^n \frac{\sum_i^m \left( w_{i+1,j} - w_{i,j} \right)}{(m-1)} / n$$
$$\frac{\partial w}{\partial r} = \frac{1}{0.025} \sum_i^m \frac{\sum_j^n \left( w_{i,j+1} - w_{i,j} \right)}{(n-1)} / m$$

## Genomic sequencing and mutation identification

Genomic DNA of isolated clones or mixed populations were extracted using Genomic DNA Extraction Kit (ThermoFisher), libraries were prepared using Nextera Kits (illumina) and sequenced by illumina HiSeq 2000 (Single-read. 50 cycles) with 160×coverage. Genomes were assembled and mutations identified using the Breseq pipeline with the wildtype strain MG1655 serving as the control.

## Structural modeling of the fliA mutation

Although crystal structures of FliA bound to DNA are not available, clues to the functional role of arginine 220 can be deduced from its location in the conserved helix-turn-helix motif for DNA binding (*Meinhart et al., 2003*). Indeed, this conserved arginine is critical to DNA binding through electrostatic interactions and hydrogen bond formation with the backbone phosphate of DNA in the homologous *Thermus aquaticus* sigma factor A (*Campbell et al., 2002*). Modeling the R220W mutation into sigma factor A eliminates the electrostatic interaction and the hydrogen bond to the backbone phosphate of DNA.

## Genetic engineering

Of the four mutations identified in the phase 3 isolate (P3), *fliA* was chosen for further study. Four strains were constructed: the ancestral background with the ancestral *fliA*, the ancestral background with the P3*fliA*, the P3 background with ancestral *fliA*, and the P3 background with the P3*fliA*. 1 Kb sequences upstream and downstream of the promoter of the *dcyD* operon (adjacent to the*fliA* operon)were PCR amplified separately. The kanamycin resistance cassette from pKD13[S] was PCR amplified and inserted between the two chromosomally derived sequences by fusion PCR[S]. The kanamycin resistance cassette was integrated between the terminator of *fliA* operon and the promoter of *dcyD* operon by lambda red-mediated homologous recombination. Single colonies with the kanamycin resistance cassette were isolated and stored at −80°C. P1 phage transduction[S] was then used to introduce both ancestral and P3 versions of *fliA* into the P3 and ancestral backgrounds respectively. The four reconstructed strains were verified by Sanger sequencing the *fliA* locus

(Biomedical Genomic Center of University of Minnesota). The constructs for ectopic promoter analysis monitoring c-di-GMP and *fliC* expression were made as follows. To monitor c-di-GMPa 1 Kb region upstream of *lacI*, kanamycin resistance cassette, *tac* promoter, Vc2 riboswitch, *gfp*, and a 1 Kb region downstream of *lacA* were fused by PCR[S]. To monitor *fliC* expression a 1 Kb region upstream of *lacI*, kanamycin resistance cassette, *fliC* promoter, *gfp*, and 1 Kb downstream of *lacA* were fused by PCR[S]. The synthetic circuits were integrated into appropriate host genome using lambda red recombination[S] with concomitant loss of *lac*. Source templates: Vc2 from *Vibriocholerae* O395N1 toxT::lacZ; *pfliC*, 1 Kb regions upstream of *lacI* and downstream of *lacA* from *E. coli* MG1655; *tac* promoter by chemical synthesis; kanamycin cassette from plasmid pKD13[S].

## Acknowledgements

The authors thank M Travisano, RF Denison, D Drabeck, WH Ratcliff and S Wan for many comments on previous versions of the manuscript.

## Additional information

### Funding

| Funder | Grant reference number | Author |
| --- | --- | --- |
| University of Minnesota | Research Support | Xiao Yi |

The funders had no role in study design, data collection and interpretation, or the decision to submit the work for publication.

### Author contributions

XY, Conception and design, Acquisition of data, Analysis and interpretation of data, Drafting or revising the article; AMD, Drafting or revising the article, Contributed unpublished essential data or reagents

### Author ORCIDs

Xiao Yi, http://orcid.org/0000-0003-4025-856X
Antony M Dean, http://orcid.org/0000-0002-9546-7679

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

# Appendix

## Model validation

Two assumptions inour model of the adaptive landscape are

1. Strains compete for common resources in the absence of other interactions.

2. Growth in TB is logistic and strains have the same carrying capacity.

Previous work has demonstrated that laboratory strains of *E. coli* grow in exactly the same way in mixed culture as they do in monoculture (*Yi and Dean, 2013*). Known exceptions are: (i) growth in minimal media with excess glucose can result in cross-feeding with acetate and glycerol metabolites in the presence of an acetate specialist[S] (*Treves et al., 1998*); (ii) a small fraction of a population sacrifices itself to promote resistance to antibiotics (*Lee et al., 2010*). Our experimental system does support the conditions for these interactions.

The logistic equation provides a reasonable fit to the growth data (*Figure 2—figure supplement 6*). Importantly, the decelerating growth after hour 6, which is responsible for the curvature of the landscape surface, is accurately captured by the logistic equation. We will later show that decelerating growth introduces curvature to the landscape resulting in an optimum at the Pareto front of the trade-off.

Carrying capacities vary three-fold among the isolates. We introduce a scalar $k$ to change the mutant carrying capacity. This gives:

$$\frac{dm}{dt} = r_m m (1 - km - w)$$
$$\frac{dw}{dt} = r_w w (k - m - w)$$

*Figure 2—figure supplement 7a* shows that both $r$ and $k$ contribute to selection during growth, $s_g$. Within the range of values for r and k seen among isolated clones $s_g$ is dominated by changes in $r$ while changes in $k$ make little difference (*Figure 2—figure supplement 7b*). The modulation of *r* 's effect by $k$ is limited for most isolates and is significant only for phase 3 isolates with large growth rates. We conclude our model is robust to assumptions 1 and 2.

