## [Decision Letter]

[Editors’ note: a previous version of this study was rejected after peer review, but the authors submitted for reconsideration. The first decision letter after peer review is shown below.]

Thank you for submitting your work entitled "Phenotypic plasticity as an adaptation to a functional trade-off" for consideration by *eLife*. Your article has been reviewed by four peer reviewers (including the Reviewing Editor) and Ian Baldwin as the Senior Editor.

Our decision has been reached after consultation between the reviewers. Based on these discussions and the individual reviews below, we feel that the message is interesting, but it may take you more than 2 months to fix all problems. Our main concerns are in how you performed and interpreted experiments, and in the mathematical model. Thus, we regret to inform you that we reject your submission at its current stage. However, if you can address all our concerns, *eLife* will be happy to consider a re-submission.

*Reviewer #1:*

The manuscript by Yi and Dean showed that phenotypic plasticity can evolve to break a functional trade-off between growth and chemotaxis. The manuscript started with a model prediction on how bacteria exposed to fluctuating environment and selection (selecting for growth following by selecting for chemotaxis) should evolve. Indeed, such prediction was shown to hold for early stage of evolution where cells initially evolved to be better at both growth and chemotaxis (my interpretation here differs somewhat from that of authors) and at phase 2 evolved along a Pareto front as cells evolved to be slower grower and faster swimmer. At phase 3 (P3), a single mutation in fliA allowed cells to break this tradeoff. *fliA* mutant lowered swimming speed during exponential growth when chemotaxis is not selected for, and increased swimming speed and fraction of motile cells at high cell density. The results are interesting. However, major issued will need to be addressed before acceptance.

1) I am not sure that I understand the lack of reproducibility across figures. Figure 2 shows that ancestor at 0 hr has a swimming speed of 8 μm/s. In Figure 3, fraction motile in ancestor is close to 0, and the swimming speed is only ~2 8 μm/s (Figure 3). This type of several-fold difference makes me suspicious.

2) Even though mutation in filA does seem to have an effect since fliA mutation is sufficient to increase swimming speed and% motile cells during late stage growth (Figure 3), some other mutation(s) in P3 can also yield the same phenotype (compare late stage growth of solid red in Figure 3 with sold black in Figure 3, and solid red and solid black in Figure 3). My hypothesis is further supposed by the observation that in Figure 2, the red line seems to show a much more drastic phenotype during early stages than the *fliA* mutant.

3) The writing is often confusing. For example, Figure 1 makes one believe that they are doing experiments, but Figure 1 give the impression of modeling. They do not lay out their model clearly (especially in the fifth paragraph of the main text where chemotaxis enters the game). Note that fitness (growth rate) can be quantified since it is constant during exponential phase. However, for chemotaxis, does population ratio change exponentially over time? In other words, would you get very different results if the duration of chemotaxis assay varies?

4) I am not sure of the usefulness of the interpretation that "usually without significant changes in growth rate" (main text, sixth paragraph) during the initial stage of evolution. I think that averaging across clones is not a good way of quantifying. If you have two populations of say 6 individuals, one with and one without a fast-growing individual, statistically speaking the two populations are the same. But the population with fast-growing individual will quickly deviate from the population without fast-growing individual. Why not just plot all six clones that you quantified, each with an error bar for measurement? My interpretation is that some clones must have significantly improved growth rate to generate the data of Figure 2. This will also make later statements (i.e. decrease in growth rate) more sensible, since most clones in later stages do not seem to have smaller growth rate than the ancestor.

5) How might one refute the alternative hypothesis that fliA is pleiotropic and that growth rate improvement has nothing to do with swimming rate plasticity? They seem to need a control where they destroy plasticity, and show that the growth advantage goes away. If this control is difficult to perform, you will at least need to acknowledge the existence of this possibility and discuss why you cannot perform such an experiment and why you think that this possibility is unlikely.

*Reviewer #2:*

Yi and Dean bring molecular genetics to bear on the problem of complex adaptation involving a functional trade-off. Some general aspects of the process have been recognized in a qualitative manner, but this study provides interesting quantitative evidence on the evolutionary trajectory followed by populations of *E. coli* subjected to alternating selection for rapid growth and chemotaxis. They used serial transfer of innocula from populations that increased from a lag phase, through an exponential growth phase, to stationary phase which they simply called "growth rate". The transfer was carried out by allowing cells to swim into a capillary for 30 minutes before being deposited in fresh medium. The efficiency of transfer was considered a measure of "chemotaxis" and they directly showed swimming speed, measured from video micrographs, correlated with the number of bacteria that entered into the capillaries.

Five separate populations were transferred twice daily for 9 weeks. They found that during the first 6 weeks chemotaxis improved in the populations but that there was a trade-off in growth rate such that it decreased (Figure 2 and Figure 2—figure supplement 1). During the next few weeks chemotaxis stayed high and the growth rate improved until it exceeded that of the original population. This is a classical sequential adaptation of traits overcoming a trade-off where initially the rate of swimming increased but the energy expended in rapid swimming limited the growth rate; further evolution resulted in reduced rate of swimming in exponentially growing cells with a return to rapid swimming just before they were presented with a capillary for transfer. The growth rate of these more highly evolved populations exceeded that of the original population. These results are clearly and thoroughly presented.

Yi and Dean then sequenced representative isolates from week 7, when only chemotaxis had improved, and week 9, when chemotaxis was still high but the growth rate had also increased. At week 7 they found mutations in the upstream region of the gene encoding the enzyme that synthesizes c-di-GMP (yegE) and in the gene encoding the c-di-GMP phosphodiesterase (yahA). The level of c-di-GMP was much reduced in low density cultures of these cells (Figure 2). This may account for the increase in chemotaxis of these cells since c-di-GMP is a signal that negatively modulates flagellar activity. There were 3 other mutations in the population but they were not analyzed and may have been irrelevant. It would be interesting to know if these same mutations were found in the other 4 populations carried through the selection. Genes that were hit two or more times are likely to be pertinent to the increase in fitness. The mutation in yahA appears to increase the level of phosphodiesterase and so is likely to be a gain-of-function dominant mutation. The authors should consider establishing this point.

The population from week 9 was found to carry a mutation in fliA which encodes the sigma factor that regulates flagellar operons. Reduction in this sigma factor may account for the decrease in the rate of swimming during early stages in the growth cycle. The subsequent increase in swimming rate might result from the utilization of a separate sigma factor for the flagellar operons as a quorum is reached. It was not clear in Table 1 whether all of the mutations seen at week 7 were still present in week 9 along with the three new mutations. This should be clarified.

Expression from the FliA-dependent fliC promoter was studied in cells from the week 9 population (P3) that were transformed with either wild type or mutant fliA (Figure 3). The wild type allele reversed the phenotype showing that the fliA mutation is recessive. The mutant allele shows that the rapid increase in expression of fliC and increase in swimming rate that occurs as the population enters the stationary stage is FliA independent. It seems possible that another sigma factor regulates flagellar operons during late stages of the growth cycle. The authors did not mention such a possibility and might want to comment on it.

The authors state that the differences in the fraction of motile cells are statistically significant but the differences are not great. They might consider omitting this point.

It seems that the rate of swimming had to increase before a change in the pattern of expression of flagellar genes could allow an increase in growth rate. If the regulatory mutation in the sigma factor happened first, the cells might not have been able to swim fast enough to compete in the capillary until the c-di-GMP mutations arose. A discussion along these lines might be useful.

This is an interesting study of the evolutionary response of an *E. coli* population to a complex selection regime. However, the results rest on genomic analyses from a single population and, as such, are not much more than anecdotal. It would be necessary to have genomic analyses from isolates of two or more populations that had undergone the exact same selection to be certain that the evolutionary trajectory had been recognized. Also, nowhere in this manuscript did the authors show that movement of bacteria into the capillary is chemotactic. They should stick with speed of swimming which is what they measure. In fact, I doubt that *E. coli* in stationary state in rich media is chemotactic to fresh rich media.

*Reviewer #3:*

This work explores a very interesting plastic response of *E. coli* to environment. I'm unable to complete the review at this stage, however, until the authors address the questions below.

I don't understand the measured correlation between swimming speed and chemotactic ability. The evidence is presented in Figure 2—figure supplement 4. First, there are no units. Second, please include error bars for each data point. Third, if difference in chemotaxis is strongly correlated with swimming speed, why the difference in chemotaxis for the highest yellow data point is ~1.8 while the difference in chemotaxis for the lowest blue data point is ~0.5, as I marked by arrows? These two clones have very similar swimming speed.

Also, I understand that measurement of the swimming speed and the difference in chemotaxis is performed in the phosphate buffer (subsections “Strains and media” and “Measuring growth rates and chemotactic abilities”) while in the propagation experiment the capillary is inserted directly into the culture (Figure 1 and caption). If my understanding is correct, the measured difference in chemotaxis does not reflect the chemotactic ability of the bacteria that get into the capillary during propagation. First, *E. coli*'s swimming speed depends on pH of the medium because its flagellar motor is powered by protons. However, for unbuffered rich medium, such as LB, which is similar to tryptone broth, the pH reaches 9 after 1day of growth (J. Bacteriol. December 2007, vol. 189, 8746-8749). Have you measured the swimming speed in the growth medium at the end of the 11.5 hours? Second, *E. coli*'s chemotactic response to the same chemical cue is very different depending on the environment it has adapted to, as demonstrated by the classical papers by J. Adler. Third, during chemotaxis, *E. coli* executes the run-tumble motility pattern so its speed is not constant. By swimming speed do you mean the instantaneous speed or average speed?

*Reviewer #4:*

I think this is a high quality paper that will appeal across several disciplines. I think its main problem is that it is not readily accessible to the audience that it will otherwise attract. None of my comments is fatal nor requires additional empirical work, but I think the comments warrant a serious overhaul or at least attention to that possibility.

1) Perspective. The paper is in essence a study of a population (strain) evolving to escape an apparent trade-off. The perspective could be elaborated more fully. For example, I can imagine two different scenarios for evolution to escape a trade-off: (a) a trade-off exists because of a true genetic constraint, and the population eventually evolves to change that constraint, allowing new phenotypes to evolve that are no longer constrained in the original way; (b) an apparent trade-off exists because all common variation/mutations obey the trade-off, but rare mutations sometimes violate the trade-off. My sense is that the latter is operating in this paper, because the R220W mutation is stated to function in a wild-type background the same as it does in the background in which it evolved. Thus (I imagine), the mutation would have been favored initially had it arisen then. (But maybe it would only have been favored in the new background?)

Regardless of how the authors view their system, I think that some presentation of a perspective on trade-offs is warranted, especially since their genetic dissection is so deep. The novelty of the paper is the genetic dissection. I also think that there are many examples that can be cited as cases of evolution of new constraint functions. Lenski's citrate metabolism is one (which was apparently Barry Hall's discovery), Hall's eBG system, and no doubt some of Tony's work would qualify. It will help to unite this work with more of the literature.

2) The mathematical model. I have some issues with or misunderstanding of the model. Growth rate clearly fits an ODE-based model of fitness, as in their (unnumbered) equations. The fit of the 'chemotaxis' part is not obvious. Here, a small tube is plunged into a volume of cells (how much?), and at 30 minutes is hauled out, the contents of the tube being used to start the next generation.

I am guessing that this is a very different process than subjecting the entire population to a common environment in which small differences in behavior lead to quantitative increases in success for all members of the population that express that behavior. However, I confess to not knowing how chemotactic selection was carried out. Methods describe a procedure for evaluating chemotaxis rate, but it is not clear that the volumes used in that assay are the same used in the selection. (180 μl seems a small volume to use for an 11.5hr growth.)

To the extent I imagine it then, the chemotaxis selection is something more akin to truncation selection with a probabilistic success rate because cells that don't end up in the capillary tube have 0 fitness even if they chemotax at some rate. Furthermore, the nature of selection is not the same for all cells in the culture exposed to the capillary tube, because some cells will start near the capillary orifice, others much farther away. It may well be that the selection can be approximated the way they have done, but I want to see some kind of argument or derivation. Normally, I would not fuss, but the paper claims that their trade-off functions are derived a priori, which is not compatible with what we have been given. And maybe this criticism stems from my inability to grasp the protocol details.

I also do not see how the trade-off surface in Figure 2 is derived. Here, there are further assumptions about how *c_m_ – c_w_* trades off with *r_m_ – r_w_.* The paper argues up front that the physiological basis of the trade-off is energy, but how is the energetic equivalence between chemotaxis and growth calculated – in the context of this selection? (It might make more sense to use standard notation such as delta for the incremental difference of the mutant from wild-type.) And perhaps a related point is that I do not understand why the logistic growth is (apparently) claimed to reduce selection for faster growth once the population has evolved to the point that it saturates in 11.5 hours – isn't there still selection for even faster growth, because any faster mutant will overtake the others? I'm clearly not following things here.

If some of these criticisms are foundless, I hope they at least help the authors recognize the bases of my confusion.

[Editors’ note: what now follows is the decision letter after the authors submitted for further consideration.]

Thank you for submitting your article "Phenotypic plasticity as an adaptation to a functional trade-off" for consideration by *eLife*. Your article has been favorably evaluated by Naama Barkai (Senior Editor) and two reviewers, one of whom is a member of our Board of Reviewing Editors. The reviewers have opted to remain anonymous.

The reviewers have discussed the reviews with one another and the Reviewing Editor has drafted this decision to help you prepare a revised submission.

Summary:

The author has addressed many of the comments raised by reviewers. However, there are still key issues that will need to be addressed.

Essential revisions:

In the subsection “Measuring growth rates and chemotactic abilities”: the culture used for measurements was grown at 37 Celsius degrees. Is this a typo? If not, the measured swimming speed and growth rates don't reflect those when the cells were grown at 30 Celsius degrees in the evolution experiments.

It is necessary to establish that the swimming speed and fraction of motile cells to be similar in phosphate buffer as in growth medium TB during the first 11.5 hours. This is because *E. coli*'s swimming speed, and thus the energy cost, depends on pH and TB is not buffered.

Plot error bars using the same color as the average. The error bars in Figure 2, especially for ancestor at time zero (which I had reservations), are difficult to see. You can always shift the data points slightly in the x-direction so that error bars do not overlap.

Your Abstract is still confusing. For example, "growth close to carrying capacity": this made me think that you used a turbidostat to grow cells at near saturation. However, in reality, your cells experienced a substantial period of exponential growth (8hrs out of 11.5). Or did I actually miss something? Also, please excuse the slowness of this reviewer. I must confess that I still have no idea what your last sentence means. For example, pleiotropy in what? You are probably aware of the different definitions of evolvability. Thus, some of these "commonly used terms" can be confusing unless you define them rigorously.

Figures are generally of poor quality (especially axis marking). Figure fonts and symbols often look blurred. The authors will need to fix these problems.

---

## [Author Response]

[Editors’ note: the author responses to the first round of peer review follow.]

*Our decision has been reached after consultation between the reviewers. Based on these discussions and the individual reviews below, we feel that the message is interesting, but it may take you more than 2 months to fix all problems. Our main concerns are in how you performed and interpreted experiments, and in the mathematical model. Thus, we regret to inform you that we reject your submission at its current stage. However, if you can address all our concerns, eLife will be happy to consider a re-submission.*

*Reviewer #1:*

*The manuscript by Yi and Dean showed that phenotypic plasticity can evolve to break a functional trade-off between growth and chemotaxis. The manuscript started with a model prediction on how bacteria exposed to fluctuating environment and selection (selecting for growth following by selecting for chemotaxis) should evolve. Indeed, such prediction was shown to hold for early stage of evolution where cells initially evolved to be better at both growth and chemotaxis (my interpretation here differs somewhat from that of authors) and at phase 2 evolved along a Pareto front as cells evolved to be slower grower and faster swimmer. At phase 3 (P3), a single mutation in fliA allowed cells to break this tradeoff. fliA mutant lowered swimming speed during exponential growth when chemotaxis is not selected for, and increased swimming speed and fraction of motile cells at high cell density. The results are interesting. However, major issued will need to be addressed before acceptance.*

*1) I am not sure that I understand the lack of reproducibility across figures. Figure 2 shows that ancestor at 0 hr has a swimming speed of 8 μm/s. In Figure 3, fraction motile in ancestor is close to 0, and the swimming speed is only ~2 μm/s (Figure 3). This type of several-fold difference makes me suspicious.*

In Figure 2, each curve was the speed kinetics of a single strain from a single sampling, each 300-400 cells. We chose the curves with most dramatic patterns for illustration. This concerned our reviewers, so we now plot the averages of five experimental replications. We are aware that the kinetic curves from the same strain have large variability. This is expected because we are dealing with an organismal behavior; for example, large variability in swimming speed is evident in Figure 5 of (Boehm et al., 2010). We confirmed the kinetic behaviors of different strains using rigorous statistical tests and using knock- in/out genetics.

*2) Even though mutation in fliA does seem to have an effect since fliA mutation is sufficient to increase swimming speed and% motile cells during late stage growth (Figure 3), some other mutation(s) in P3 can also yield the same phenotype (compare late stage growth of solid red in Figure 3 with sold black in Figure 3, and solid red and solid black in Figure 3). My hypothesis is further supposed by the observation that in Figure 2, the red line seems to show a much more drastic phenotype during early stages than the fliA mutant.*

We agree. As mentioned in the eleventh paragraph of the main text, any or all of three other mutations in the focal evolvant also increase the fraction of motile cells, and thus the average swimming speed in late hours. This does nothing to undermine our main point which is that the fliA mutation reduces swimming speed in early hours and increases speed in late hours. Put it another way, the dual effect of fliAon swimming speed is still operational in P3 background at an elevated baseline boosted by the three other mutations.

*3) The writing is often confusing. For example, Figure 1 makes one believe that they are doing experiments, but Figure 1 give the impression of modeling. They do not lay out their model clearly (especially in the fifth paragraph of the main text where chemotaxis enters the game). Note that fitness (growth rate) can be quantified since it is constant during exponential phase. However, for chemotaxis, does population ratio change exponentially over time? In other words, would you get very different results if the duration of chemotaxis assay varies?*

The modeling of the adaptive landscape was intended to describe the experiments and provide a guide to predict the future trajectory of phenotypic evolution. To clarify this point, we add subheadings 'a. Experimental design' and 'b.Predicted adaptive landscape. The experimental system defines an adaptive landscape that can be described mathematically from first principles.'

In the resubmission, we add a reference to the classic paper by Adler (Adler, 1973) that shows the number of cells entering a capillary carrying attractant increases exponentially in the first 30 min of incubation (Adler, 1973 Figure 2). Hence, both growth and capillary selection are exponential in nature. They are therefore conceptually equivalent. We fixed the duration of chemotaxis to be 30 min throughout the experiments and thus the selection due to chemotaxis can be represented as a simple difference. Also, 30 mins sufficiently short that fewer than 200,000 cells enter the capillary. Hence, saturation is of no concern.

*4) I am not sure of the usefulness of the interpretation that "usually without significant changes in growth rate" (main text, sixth paragraph) during the initial stage of evolution. I think that averaging across clones is not a good way of quantifying. If you have two populations of say 6 individuals, one with and one without a fast-growing individual, statistically speaking the two populations are the same. But the population with fast-growing individual will quickly deviate from the population without fast-growing individual. Why not just plot all six clones that you quantified, each with an error bar for measurement? My interpretation is that some clones must have significantly improved growth rate to generate the data of Figure 2. This will also make later statements (i.e. decrease in growth rate) more sensible, since most clones in later stages do not seem to have smaller growth rate than the ancestor.*

We replaced the sentence with " few isolates have changed growth rates". We absolutely agree with the reviewer that the isolates can vary from each other in phenotype. Averaging across the isolates allows the overall the population trajectory to be clearly mapped onto the adaptive landscape, emphasizing the overall tempo and mode of phenotypic evolution. Data for individual isolates are presented in Figure 2—figure supplement 3 for interested readers.

*5) How might one refute the alternative hypothesis that fliA is pleiotropic and that growth rate improvement has nothing to do with swimming rate plasticity? They seem to need a control where they destroy plasticity, and show that the growth advantage goes away. If this control is difficult to perform, you will at least need to acknowledge the existence of this possibility and discuss why you cannot perform such an experiment and why you think that this possibility is unlikely.*

The alternative hypothesis, that the fliA mutation is pleiotropic and that growth rate improvement has nothing to do with swimming rate plasticity, requires yet another hypothesis to explain why fliA mediated swimming rate plasticity should no longer improve growth rate in the face of a demonstrated trade-off. One possibility is that the fliA mutation itself somehow negates the hard trade-off between energy for growth and energy for chemotaxis, although the means by which this might be achieved is highly speculative. We feel our interpretation is less convoluted – the fliA mutation alters swimming rate in the manner expected to increase fitness.

Two independent lines of evidence support causality between growth rate reduction and swimming speed increase. The first comes from Figure 3. Introducing the fliA mutation to the ancestor increases the growth rate while decreasing the swimming speed in the early hours. The second comes from the empirical observation that, from week 2 to 7, the populations had decreased growth rates while chemotaxis increased (Figure 2). This trend was not explained by the fliA mutation since this mutation was only isolated from week 9. For instance, the representative strain of week 7 did not carry it (Table 1). Hence, the trade-off between growth and chemotaxis is not fliA-dependent, suggesting it is due to a more fundamental mechanism such as the hard trade-off between energy for growth and energy for chemotaxis. These results are entirely consistent with the hypothesis that the fliA mutation alters swimming rate in the manner expected to increase fitness.

Reviewer #2:

*[…] Yi and Dean then sequenced representative isolates from week 7, when only chemotaxis had improved, and week 9, when chemotaxis was still high but the growth rate had also increased. At week 7 they found mutations in the upstream region of the gene encoding the enzyme that synthesizes c-di-GMP (yegE) and in the gene encoding the c-di-GMP phosphodiesterase (yahA). The level of c-di-GMP was much reduced in low density cultures of these cells (Figure 2). This may account for the increase in chemotaxis of these cells since c-di-GMP is a signal that negatively modulates flagellar activity. There were 3 other mutations in the population but they were not analyzed and may have been irrelevant. It would be interesting to know if these same mutations were found in the other 4 populations carried through the selection. Genes that were hit two or more times are likely to be pertinent to the increase in fitness. The mutation in yahA appears to increase the level of phosphodiesterase and so is likely to be a gain-of-function dominant mutation. The authors should consider establishing this point.*

We appreciate the reviewer's interest and suggestion in delving into the mechanisms underlying the "brute force" increase of swimming speed seen in the representative strain of week 7. Our focus in the current paper is how the growth-chemotaxis trade-off is alleviated by a new behavior seen in the strain isolated from week 9. To avoid distraction, we think it better to save the work on week 7 isolates for another study.

*The population from week 9 was found to carry a mutation in fliA which encodes the sigma factor that regulates flagellar operons. Reduction in this sigma factor may account for the decrease in the rate of swimming during early stages in the growth cycle. The subsequent increase in swimming rate might result from the utilization of a separate sigma factor for the flagellar operons as a quorum is reached. It was not clear in Table 1 whether all of the mutations seen at week 7 were still present in week 9 along with the three new mutations. This should be clarified.*

We have interpreted our results based on the decades of work by multiple laboratories that have thoroughly characterized chemotaxis in *E. coli*. There is no evidence that *E. coli* flagellar operons are regulated by sigma factors other than fliA and sigma70. There is no evidence that *E. coli* flagellar operons are regulated by quorum sensing. There is no compelling reason to pursue these hypotheses given that our explanations are currently sufficient. We specified in the caption to Table 1: “Note that there is no overlap in mutations between the two isolates”.

*Expression from the FliA-dependent fliC promoter was studied in cells from the week 9 population (P3) that were transformed with either wild type or mutant fliA (Figure 3). The wild type allele reversed the phenotype showing that the fliA mutation is recessive. The mutant allele shows that the rapid increase in expression of fliC and increase in swimming rate that occurs as the population enters the stationary stage is FliA independent. It seems possible that another sigma factor regulates flagellar operons during late stages of the growth cycle. The authors did not mention such a possibility and might want to comment on it.*

We do not understand how the reviewer concludes that the changes in swimming speed are FliA independent. The fliA mutation clearly reduces swimming speed early in the growth cycle relative to wildtype fliA, just as it increases it late in the cycle relative to wildtype fliA, in both the ancestral and the evolved genetic backgrounds. We are not aware of any other sigma factors that regulate chemotaxis genes and motility besides sigma factor 28 (fliA) and sigma factor 70. It is hard to imagine how introducing the fliA mutation would cause a change of action in sigma factor 70 in late hours. At the same time, variability in the fraction of motile cells explains population-average swimming speed and/or chemotaxis. We add a new figure (Figure 3—figure supplement 2) in the resubmission to support this statement. *R*_2_ for the regression between motile fraction and swimming speed is at least 0.86.

*It seems that the rate of swimming had to increase before a change in the pattern of expression of flagellar genes could allow an increase in growth rate. If the regulatory mutation in the sigma factor happened first, the cells might not have been able to swim fast enough to compete in the capillary until the c-di-GMP mutations arose. A discussion along these lines might be useful.*

Again, isolates from week 7 and 9 do not have mutational overlap. Thus, the c-di-GMP mechanism is irrelevant to the discussion of the changed regulation of swimming speed by the fliA mutation.

*This is an interesting study of the evolutionary response of an E. coli population to a complex selection regime. However, the results rest on genomic analyses from a single population and, as such, are not much more than anecdotal. It would be necessary to have genomic analyses from isolates of two or more populations that had undergone the exact same selection to be certain that the evolutionary trajectory had been recognized.*

We evolved five independent and parallel populations and all had similar evolutionary trajectories: the populations initially evolved to the Pareto font and then broke through to reach high growth and high chemotaxis. We add: "All five parallel shared similar patterns (Figure 2—figure supplement 1)”. In this study, we focus on one population in its genetic and mechanistic dissection as an in-depth case study. Therefore, at the phenotypic level, the phenomenon of trade-off alleviation is reproducible and thus general. At the molecular/genetic level, the mechanism we identified may not be general. Evolutionary history consists of idiosyncratic events: the symbiotic event that gave rise to mitochondria and eukaryotes happened once in three billion years; rubisco, arguably the most abundant protein on the planet, had a single origin. We believe our case study has its own merit even without generality in the underlying molecular/genetic mechanisms. We are *not* making a case for generality. We are exploring how phenotypic plasticity can mitigate a trade-off.

*Also, nowhere in this manuscript did the authors show that movement of bacteria into the capillary is chemotactic. They should stick with speed of swimming which is what they measure. In fact, I doubt that E. coli in stationary state in rich media is chemotactic to fresh rich media.*

We add Figure 2—figure supplement 5 in our resubmission to show that *E. coli* cells did chemotax in our experimental conditions with a negative control using blank buffer as the attractant.

*Reviewer #3:*

This work explores a very interesting plastic response of E. coli to environment. I'm unable to complete the review at this stage, however, until the authors address the questions below.

*I don't understand the measured correlation between swimming speed and chemotactic ability. The evidence is presented in Figure 2—figure supplement 4. First, there are no units. Second, please include error bars for each data point. Third, if difference in chemotaxis is strongly correlated with swimming speed, why the difference in chemotaxis for the highest yellow data point is ~1.8 while the difference in chemotaxis for the lowest blue data point is ~0.5, as I marked by arrows? These two clones have very similar swimming speed.*

We apologize for the inappropriately prepared supplemental figures. We have now updated them in the resubmission according to the reviewer's comments. Error bars have been added to Figure 2—figure supplement 4. Note that data points of different colors were taken at different stages of growth and that the difference in chemotaxis measures the relative ability of the isolates with respect to a common competitor strain (ancestor) to be attracted into the capillary. The isolates and their corresponding competitors were sampled from the same stage of growth. Therefore, it is not meaningful to compare data points across different colors, i.e., different growth stages. But regression of data points from the same growth stage shows a good correlation between swimming speed and chemotaxis.

*Also, I understand that measurement of the swimming speed and the difference in chemotaxis is performed in the phosphate buffer (subsections “Strains and media” and “Measuring growth rates and chemotactic abilities”) while in the propagation experiment the capillary is inserted directly into the culture (Figure 1 and caption).*

Again, I apologize for the inaccurate information we provided on the experimental protocols. We indeed washed and diluted the culture in phosphate buffer and then inserted a capillary into this treated sample. We provided this information in the supplemental info for pre-submission (*eLife* has the record) and accidentally deleted it when re-arranging and shortening texts for full submission. We added the full description in the "Experimental evolution" section of the Methods.

*If my understanding is correct, the measured difference in chemotaxis does not reflect the chemotactic ability of the bacteria that get into the capillary during propagation. First, E. coli's swimming speed depends on pH of the medium because its flagellar motor is powered by protons. However, for unbuffered rich medium, such as LB, which is similar to tryptone broth, the pH reaches 9 after 1day of growth (J. Bacteriol. December 2007, vol. 189, 8746-8749). Have you measured the swimming speed in the growth medium at the end of the 11.5 hours?*

Yes, we did measure swimming speed at 11.5 hours as shown in Figure 2—figure supplement 4, and Figure 3. Note that before all speed measurements and capillary assays/selection, cells sampled from the population grown in tryptone broth were treated with a modified Adler (1973) protocol with washing and dilution in phosphate buffer (described in Methods). Therefore, chemotaxis was always carried out in a well-buffered standardized environment.

We did check the motility at 11.5h for cells sampled from tryptone broth cultures without washing. They were motile. Note that at 11.5 hours at 30°C, the cultures were transitioning from log phase to stationary phase (see the growth curve in Figure 2). They were still growing, if only slowly, and serious complications in stationary phase, such as extreme pH and secondary metabolites, would still be minimal.

*Second, E. coli's chemotactic response to the same chemical cue is very different depending on the environment it has adapted to, as demonstrated by the classical papers by J. Adler. Third, during chemotaxis, E. coli executes the run-tumble motility pattern so its speed is not constant. By swimming speed do you mean the instantaneous speed or average speed?*

Chemotaxis selection and assays were carried out under the defined standardized conditions described in the Methods. As also described in the Methods, swimming speed is calculated as the average linear velocity of a single cell over the course of 5 seconds with 16 fps. Tumbling would contribute to the slowing down of the speed.

*Reviewer #4:*

*I think this is a high quality paper that will appeal across several disciplines. I think its main problem is that it is not readily accessible to the audience that it will otherwise attract. None of my comments is fatal nor requires additional empirical work, but I think the comments warrant a serious overhaul or at least attention to that possibility.*

*1) Perspective. The paper is in essence a study of a population (strain) evolving to escape an apparent trade-off. The perspective could be elaborated more fully. For example, I can imagine two different scenarios for evolution to escape a trade-off: (a) a trade-off exists because of a true genetic constraint, and the population eventually evolves to change that constraint, allowing new phenotypes to evolve that are no longer constrained in the original way; (b) an apparent trade-off exists because all common variation/mutations obey the trade-off, but rare mutations sometimes violate the trade-off. My sense is that the latter is operating in this paper, because the R220W mutation is stated to function in a wild-type background the same as it does in the background in which it evolved. Thus (I imagine), the mutation would have been favored initially had it arisen then. (But maybe it would only have been favored in the new background?)*

We have been struggling with the conceptual framework. The current one is the result of compromise. In the resubmission, we add discussion of these two ways of overcoming trade-offs. While there have been examples of breaking the constraint directly such as Lenski 's citrate work, we argue that our work illustrates how a trade-off can be alleviated by a new behavior without breaking the constraint. The two of us have been all at sixes and sevens about behavior, plasticity and trade-offs, genetic, biochemical, apparent and real. We think the trade-off between energy for growth and energy for motility is hard wired and cannot be modified (energy for growth and for motility is pulled from the same pool). This is a physical constraint (the conservation of energy and matter) rather than a genetic constraint that might otherwise change. The trade-off is real rather than apparent. What the bugs do is minimize costs and maximize benefits by becoming phenotypically plastic. The trade-off is still there.

*Regardless of how the authors view their system, I think that some presentation of a perspective on trade-offs is warranted, especially since their genetic dissection is so deep. The novelty of the paper is the genetic dissection. I also think that there are many examples that can be cited as cases of evolution of new constraint functions. Lenski's citrate metabolism is one (which was apparently Barry Hall's discovery), Hall's eBG system, and no doubt some of Tony's work would qualify. It will help to unite this work with more of the literature.*

We refrained from citing an extensive body of literature in order to keep with requirements of *eLife* for short report. But we plan to write a review on adaptive landscape where a synthesis with previous work is more appropriate.

*2) The mathematical model. I have some issues with or misunderstanding of the model. Growth rate clearly fits an ODE-based model of fitness, as in their (unnumbered) equations. The fit of the 'chemotaxis' part is not obvious. Here, a small tube is plunged into a volume of cells (how much?), and at 30 minutes is hauled out, the contents of the tube being used to start the next generation.*

We apologize for the insufficient details on the experimental procedures in our previous manuscript. We did not submit the whole population (1 ml dense culture in rich medium) to capillary selection because that would saturate the capillary and the complex chemical environment in late log phase would complicate chemotaxis. So we washed and diluted cultures in phosphate buffer to a final OD600 of ~0.1- 0.2 and used 180 μl for capillary assay. We have now updated the Methods.

*I am guessing that this is a very different process than subjecting the entire population to a common environment in which small differences in behavior lead to quantitative increases in success for all members of the population that express that behavior.*

Adler showed in Figure 2 of his classic paper (Adler, 1973) that the number of cells attracted into the capillary carrying attractant increases with time in an exponential manner. Hence, both growth and capillary selection are exponential in nature. They are conceptually equivalent.

*However, I confess to not knowing how chemotactic selection was carried out. Methods describe a procedure for evaluating chemotaxis rate, but it is not clear that the volumes used in that assay are the same used in the selection. (180 μl seems a small volume to use for an 11.5hr growth.)*

The Methods have been updated to exactly describe the chemotaxis selection, which is identical to the chemotaxis assay.

*To the extent I imagine it then, the chemotaxis selection is something more akin to truncation selection with a probabilistic success rate because cells that don't end up in the capillary tube have 0 fitness even if they chemotax at some rate.*

As we have noted above, both growth and capillary selection are exponential in nature. They are conceptually equivalent. There is no truncation selection. The success rate will be almost deterministic for any mutant of appreciable frequency (say > 1%) and with a large selection coefficient (say > 1%) in the 100,000-200,000 cells entering the capillary. Our chemotaxis assays always involved mutant populations far > 1000 cells.

*Furthermore, the nature of selection is not the same for all cells in the culture exposed to the capillary tube, because some cells will start near the capillary orifice, others much farther away. It may well be that the selection can be approximated the way they have done, but I want to see some kind of argument or derivation.*

It is certain that the migration of cells into the capillary is a complex dynamic process. We have not attempted to model this process a priori. Instead, we have taken the observed exponential increase in cells in the capillary as the basis for our model. A priori prediction of the exponential parameter would akin to predicting the exponential growth rate of *E. coli* from first principles. But given that it is exponential we can construct the model.

*Normally, I would not fuss, but the paper claims that their trade-off functions are derived a priori, which is not compatible with what we have been given. And maybe this criticism stems from my inability to grasp the protocol details.*

We *never* claim the trade-off front is derived a priori. Instead, we stated in the eighth paragraph of the main text that "in our adaptive landscape the relationships between phenotypes and fitness were defined a priori". And we go on "independent regressions from different experimental populations led to similar Pareto fronts (Figure 2—figure supplement 1), with all populations evolving close to the predicted optimum (Figure 2—figure supplement 1)". Hence, the trade-off between growth and chemotaxis is entirely empirical. The Pareto font is empirical and derived by linear regression of the data points (Figure 2—figure supplement 3).

When mapping this front onto the curved adaptive landscape, which is defined a priori, a fitness optimum is derived. This provides a prediction for further experimental evolution.

*I also do not see how the trade-off surface in Figure 2 is derived. Here, there are further assumptions about how cm – cw trades off with rm – rw. The paper argues up front that the physiological basis of the trade-off is energy, but how is the energetic equivalence between chemotaxis and growth calculated – in the context of this selection? (It might make more sense to use standard notation such as delta for the incremental difference of the mutant from wild-type.) And perhaps a related point is that I do not understand why the logistic growth is (apparently) claimed to reduce selection for faster growth once the population has evolved to the point that it saturates in 11.5 hours – isn't there still selection for even faster growth, because any faster mutant will overtake the others? I'm clearly not following things here.*

The question is: Why is the adaptive landscape curved rather than a flat plane? If, say, we grow the population for only 5 hours instead of 11.5, and impose capillary selection when the cells are still in mid-log phase, then the adaptive landscape would be a flat plane. Mapping the empirical Pareto front (a straight line) onto this plane can't produce an optimum. Allowing cells to grow into late-log or early stationary phase causes a reduction in growth rates. As explained in Figure 1, with faster grow rates populations approach carrying capacity sooner. By the time capillary selection is imposed, a population with a large growth rate will be very close to carrying capacity. Further increases the growth rate will not produce much of an increase in population size. The effect of diminishing returns due to logistic growth produces curvature in the adaptive landscape.

[Editors' note: the author responses to the re-review follow.]

Essential revisions:

*In the subsection “Measuring growth rates and chemotactic abilities”: The culture used for measurements was grown at 37 Celsius degrees. Is this a typo? If not, the measured swimming speed and growth rates don't reflect those when the cells were grown at 30 Celsius degrees in the evolution experiments.*

Yes, it is a typo, now corrected.

*It is necessary to establish that the swimming speed and fraction of motile cells to be similar in phosphate buffer as in growth medium TB during the first 11.5 hours. This is because E. coli's swimming speed, and thus the energy cost, depends on pH and TB is not buffered.*

The first author did the experiment in the past month measuring swimming speed and the fraction of motile cells in the original culture and in phosphate buffer side-by-side. The result, reported in the new figure (Figure 3—figure supplement 3), shows that both measurements have similar values between the two assay conditions and that plotting one against the other gives a line with slope close to one. We agree that TB and phosphate buffer are very different and would influence chemotaxis and motility differentially.

But the measurements were taken within 5-10 min post dilution into phosphate buffer in all our experiments. We believe this interval is too brief for dramatic physiological changes to occur.

*Plot error bars using the same color as the average. The error bars in Figure 2, especially for ancestor at time zero (which I had reservations), are difficult to see. You can always shift the data points slightly in the x-direction so that error bars do not overlap.*

Error bar color is corrected. The error bar for the ancestor at time 0 in Figure 3 is smaller than the symbol.

*Your Abstract is still confusing. For example, "growth close to carrying capacity": this made me think that you used a turbidostat to grow cells at near saturation. However, in reality, your cells experienced a substantial period of exponential growth (8hrs out of 11.5). Or did I actually miss something? Also, please excuse the slowness of this reviewer. I must confess that I still have no idea what your last sentence means. For example, pleiotropy in what? You are probably aware of the different definitions of evolvability. Thus, some of these "commonly used terms" can be confusing unless you define them rigorously.*

The new manuscript reads "growth up to carrying capacity". The last sentences now state "…illustrate how phenotypic plasticity potentiates evolvability by opening up new regions of the adaptive landscape". "Opening up new regions of the adaptive landscape" annotates "evolvability".

*Figures are generally of poor quality (especially axis marking). Figure fonts and symbols often look blurred. The authors will need to fix these problems.*

We increased resolution of the pictures.